# Vaccinia virus mRNAs containing long 5'-poly(A)-leaders lack a canonical 5'-methylguanosine cap

Václav Vopálenský [1,5] ✉, Michal Sýkora [1,5], Zora Mělková[2,3], Ivan Barvík[4], Kamila Horáčková[1], Tomáš Mašek[1] & Martin Pospíšek [1] ✉

The vaccinia virus (VACV) is a prototypical poxvirus that was originally used to eradicate smallpox. Half a century ago, investigation into VACV mRNA substantially contributed to the fundamental discovery of the 5' mRNA cap, a hallmark of all eukaryotic and many viral mRNAs. VACV research also facilitated the identification and understanding of the general mechanism of 5' mRNA cap synthesis. We analyzed VACV transcripts at the level of individual mRNA molecules using a modified 5' RACE method. Our results demonstrate that VACV mRNAs containing long nontemplated 5' poly(A) leaders lack the 5' cap structure in vivo. The probability of the $m^7G$ cap occurrence decreases with the increasing number of nontemplated adenosines in the 5' poly(A) leader. Although half of VACV mRNAs with a single nontemplated adenosine still contain the $m^7G$ cap, only about 4% of viral mRNAs with leaders consisting of six or more nontemplated adenosines retain the cap. Uncapped mRNA can be transcribed from all genes containing adenosine-rich initiator sequences (INR) within their promoters. Early genes with INR still produce mostly capped transcripts (40%-59%, depending on the gene). However, intermediate mRNAs are less capped (11%-56%) and late mRNAs are mostly uncapped (0%-10% of capped mRNAs, depending on the gene).

Poxviruses belong to a family of large cytoplasmic animal and arthropod double-stranded DNA viruses, including two viruses that exclusively infect humans, namely, molluscum contagiosum virus and variola virus, the latter being the causative agent of smallpox. Before its eradication, variola virus killed more humans than all other infectious diseases collectively[1]. Some poxviruses possess the ability to cross the interspecies barrier and can be transmitted from animal hosts to humans. From this perspective, the most effective poxvirus is the monkeypox (mpox) virus, which spread rapidly in the human population worldwide as a new pandemic agent over the past three years. As of July 2025, the WHO had reported more than 150,000 laboratory-confirmed cases and 380 deaths across 137 countries[2]. Mpox infection appeared to be particularly risky for people living with HIV, for whom specific WHO guidelines have been released recently[3]. Current advances in synthetic biology and genetic engineering have led to the recent reassessment of smallpox among the most dangerous bioterrorism agents in category A, where it received the highest risk-priority score together with anthrax[4]. The characteristics of animal and human poxviruses, global travel and decreased global political stability have thus led to the inclusion of poxviruses among potential emerging agents of the next deadly pandemic. Therefore, programs aimed at developing better smallpox and mpox vaccines and treatment have

[1]Laboratory of RNA Biochemistry, Department of Genetics and Microbiology, Faculty of Science, Charles University, Prague, Czech Republic. [2]Institute of Medical Microbiology, First Faculty of Medicine, Charles University and General University Hospital in Prague, Prague, Czech Republic. [3]Biotechnology and Biomedical Center in Vestec (BIOCEV), First Faculty of Medicine, Charles University, Vestec, Czech Republic. [4]Institute of Physics, Faculty of Mathematics and Physics, Charles University, Prague, Czech Republic. [5]These authors contributed equally: Václav Vopálenský, Michal Sýkora.
✉e-mail: vaclav.vopalensky@natur.cuni.cz; martin.pospisek@natur.cuni.cz

been revitalized worldwide recently[5,6], and new data about the replication of poxviruses and their interaction with host cells and the immune system are needed.

Gene expression of vaccinia virus (VACV), a prototypical member of the *Orthopoxvirus* family used in the smallpox eradication campaign[7], proceeds as a tightly regulated cascade of sequential transcription of early, intermediate and late genes (gene time classes, GTCs) and is regulated primarily at the transcription initiation level[8]. GTCs have been reviewed simultaneously both for VACV and mpox recently[9]. VACV-encoded RNA polymerase (RNAP) is a multisubunit enzyme[10] that exists in two distinct forms that are specific for transcription of early and postreplicative genes[11]. The 5′ ends of early and postreplicative VACV transcripts are traditionally considered to be protected by the 7-methyguanosine cap structure (m$^7$G)[12,13], which is synthesized by the VACV mRNA capping enzyme[14,15]. The promoter composition and transcription regulation of VACV intermediate and late genes differ from those of early genes. Mutational analyses and transcriptome sequencing identified an essential 15-nt-long A/T-rich promoter core sequence of the VACV early genes located -12 nt upstream of the transcription start site (TSS)[16]. Promoters of VACV intermediate and late genes contain 8–11-nt A/T-rich core elements located -11 and 6 nt upstream of the initiator region (INR), respectively[17–19]. Using VACV transcriptome sequencing, consensus promoter sequences of intermediate and late genes have been determined[16,20]. Postreplicative mRNAs have relatively short 5′ untranslated regions[20], and their initiation nucleotide is adenosine both in vivo and in vitro[13,21]. Mapping of the 5′ ends of the VACV intermediate and late mRNAs by primer extension revealed 5′ poly(A) leaders ~35 nt in length that were not encoded by the viral genome[22–26]. These nontemplated 5′ poly(A) leaders are likely produced by repeated RNAP sliding on the INR in vivo and in vitro[19,27]. Transcription initiation sites for the vast majority of VACV postreplicative genes are located within 25 nt of the translation initiation codon[20]. While the exact biological role of the 5′ poly(A) leaders in postreplicative transcripts has long remained unknown, recent research has shed some light on their function. It has been demonstrated that the 5′ poly(A) leaders provide viral mRNAs with a translational advantage over the host cell transcripts in infected cells[28] and that VACV reprograms the host translation apparatus to prioritize translation of its own and a specific set of host mRNAs after the VACV-induced translational shutoff[29–31]. We showed that VACV RNAP belongs to the monophyletic group with RNAPs encoded by yeast virus-like elements (VLEs), and the VLE promoters are related to those of poxviruses[32]. VLE mRNAs also contain 5′ poly(A) leaders of variable length that are not complementary to the VLE DNA[33]. We provided evidence that 5′ poly(A) leaders of VLE mRNAs are formed by a mechanism similar to the synthesis of 5′ poly(A) leaders of the VACV postreplicative transcripts[32]. We further demonstrated that 5′ highly polyadenylated VLE mRNAs do not contain the m$^7$G cap moiety at their termini. Similarities between VACV and VLEs in terms of RNA polymerases, promoters and 5′ mRNA poly(A) leaders[32,33] prompted us to perform a detailed characterization of VACV mRNAs at the level of individual mRNA molecules.

## Results

The presence of the 5′ cap at the 5′-ends of VACV transcripts is commonly recognized, and VACV was among the first DNA viruses in which the 5′ mRNA cap moiety was identified[12,34,35]. By analysis of the total VACV early and postreplicative mRNAs synthesized in vivo, it has been estimated that the 5′ cap structure occurs once per thousand nucleotides in early transcripts and once per 1250 nucleotides in postreplicative mRNA transcripts[13]. However, the presence of the 5′ cap structure at individual VACV transcripts has never been tested. It is technically demanding to analyze the presence of m$^7$G at the individual transcripts. To our knowledge, previous analyses of m$^7$G presence at VACV mRNAs evaluated collectively all transcripts from a single gene

or group of genes or were biased by preferential selection of capped mRNAs. To achieve the goal of m$^7$G cap analysis at individual VACV mRNAs, we decided to apply a modification of the Rapid Amplification of 5′ cDNA Ends (5′ RACE) method that allows to specifically read the sequence at the 5′ mRNA ends, including m$^7$G cap detection. In brief, the method takes advantage of SuperScript III reverse transcriptase's ability to bypass the 5′–5′ triphosphate bond of the m$^7$GpppN cap and insert cytosine into the cDNA position, which is complementary to the 5′ methylguanosine cap moiety. The cDNA was then tailed using terminal deoxynucleotidyl transferase and dGTP. The 5′ mRNA ends were inferred upon amplification using the cDNA as a template, an oligo(dC) anchor primer, and a gene-specific primer. Resulting amplicons were cloned into the pCR4-TOPO vector and sequenced. To eliminate amplification bias, clones from several independent amplification rounds were used for the analysis[33]. We successfully applied this method previously for analysis of 5′ polyadenylation and capping of VLE mRNAs[32,33]. We purified total RNA from HeLa cells infected and mock-infected with VACV and investigated the 5′ ends of selected mRNAs covering all three GTCs. Typical results from the 5′ RACE analysis of VACV transcripts are depicted at Fig. 1. Early mRNAs are represented by completely capped *H5R* transcripts. This was not the case for the intermediate *G8R* gene and late *A17L* gene. From 18 transcripts analyzed for each of the *G8R* and *A17L* genes, only 6 (33%) and not even one (0%) contained m$^7$G cap at their 5′ ends, respectively. Figure 1 also demonstrates that uncapped transcripts were extended with 5′ nontemplated poly(A) leaders, while capped transcripts were not subjected to extensive 5′ polyadenylation. We extended the 5′ RACE analysis to individual transcripts of 15 genes evenly covering all three GTCs (Fig. 2, Supplementary Fig. S1–S3 and Table S2). Consistent with the published results[12,36], our analysis revealed that VACV early mRNAs, which are controlled by promoters lacking INR (represented here by the genes *J6R, K1L, H5R* and *I4L*), lacked 5′ poly(A) leaders and contained a 5′ cap. In contrast, only approximately half of the VACV early transcripts controlled by promoters containing the adenosine-rich INR, and thus bearing 5′ poly(A) leaders, contained a 5′ cap structure (represented by the genes *A5R, D5R* and *D12L*). Collectively, VACV intermediate transcripts were capped even less frequently. We identified a 5′ cap moiety at 36, 33 and 11% mRNAs of the intermediate genes *A2L, G8R* and *A1L*, respectively. Capped late VACV mRNAs were detected only sporadically (represented by the genes *C3L, L3L* and *F17R*) or not at all (*A17L*). Consistent with previous findings[37], we confirmed that the length of the 5′ poly(A) leader in VACV transcripts increases in each successive GTC (Fig. 3 and Supplementary Table S3). As in VLE mRNAs[33], the occurrence of the 5′ cap in VACV transcripts was significantly negatively correlated (*Pearson r* = − 0.72, *p* = 0.003, *N* = 306) with the increasing number of nontemplated adenosines in the 5′ poly(A) mRNA leaders (Fig. 4 and Supplementary Table S4).

Although we proved the accuracy of our 5′ RACE method and confirmed the obtained results by indirect biological and biochemical assays previously[33], we decided to test its performance directly using chemically synthetized RNA oligonucleotides containing 5′ poly(A$_{17}$) leader and either 5′ triphosphate or 5′ m$^7$G cap. Length of the 5′ poly(A$_{17}$) leader was designed according to median length of the *A17L* 5′ poly(A) leaders (Fig. 2, Supplementary Fig. S3, S4 and Supplementary Tables S2, S5). We analyzed 25 and 23 independent 5′ RACE clones for the 5′-ppp-(A$_{17}$)-RNA and 5′-m$^7$G-(A$_{17}$)-RNA oligonucleotides, respectively. In both cases, we detected a slight extension of poly(A) leaders by four adenosines in average. No m$^7$G caps were detected in the case of 5′-ppp-(A$_{17}$)-RNA, while in the case of 5′-m$^7$G-(A$_{17}$)-RNA we detected presence of 5′ caps in 18 of 23 5′ RACE clones tested (Fig. 2; Supplementary Fig. S4 and Table S5). We assume that a slight extension of 5′ poly(A) leaders measured by 5′ RACE is caused by the reverse transcriptase slippage. A few missing caps are probably caused by an incomplete modification of the RNA oligonucleotide or by hydrolytic loss of m$^7$G caps during the oligonucleotide chemical synthesis and/or

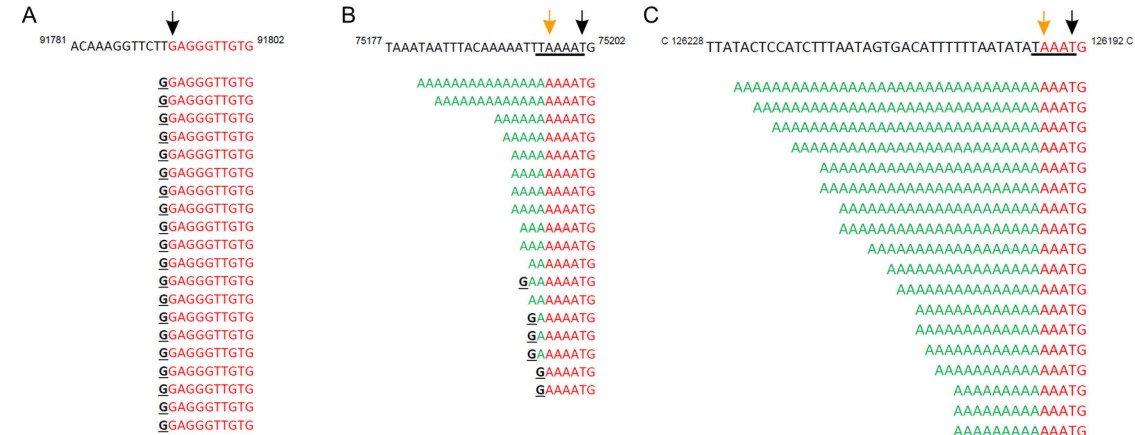

**Fig. 1 | 5′ RACE analysis of VACV transcripts. A** *H5R* early gene transcripts are all capped (see also Supplementary Fig. S1); the first 10 nucleotides of the 5′ untranslated region are displayed; the ATG translation initiation start codon is located 78 nt downstream from the TSS (Supplementary Fig. S1). **B** Only a part of the *G8R* intermediate transcript contains 5′ cap moieties (see also Supplementary Fig. S2). **C** All *A17L* late transcripts contain long 5′ poly(A) leaders and are not capped (see also Supplementary Fig. S3). The upper sequence in each panel corresponds to the viral template DNA. The TSS (black arrow) and INR (underlined) were annotated according to Yang et al. 2011 and 2012[16,37]. The sequences depicted below the template DNA represent individual sequenced cDNA clones. The 5′ untranslated regions are shown up to the ATG translation start codon, except in panel (A). Nucleotides identical to the viral template DNA are labeled in red. Nucleotides added in a nontemplated manner are labeled in green. The guanosine residues corresponding to the 5′ mRNA cap are marked in black. All sequences are shown in the 5′-to-3′ orientation, regardless of their transcriptional orientation in the VACV genome.

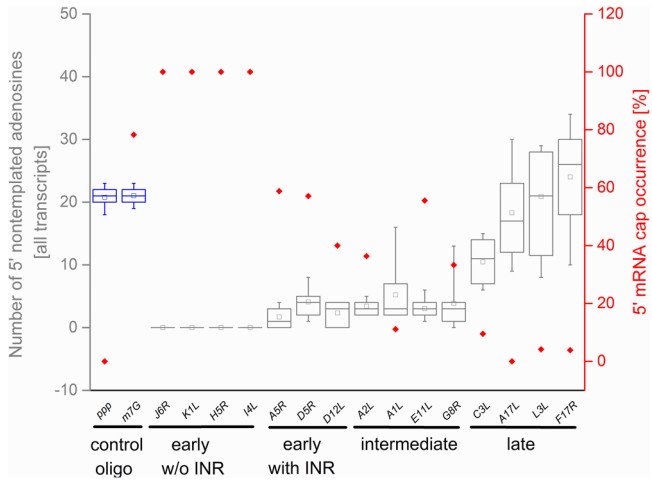

**Fig. 2 | Analysis of 5′ poly(A) leader length and cap occurrence in VACV transcripts from different GTCs.** The plot shows a negative correlation between the 5′ poly(A) leader length (left Y-axis) and cap occurrence (right Y-axis) in mRNAs from different successive GTCs represented by selected early, intermediate and late vaccinia virus genes (X-axis). The box-whisker plot (in gray) represents the lengths of nontemplated 5′ poly(A) leaders. The lower bar, upper bar, bottom and top of the boxes represent the 10th percentile, 90th percentile and the first and third quartiles, respectively. The square dot and inner line in each box represent the mean and median, respectively. The proportions of 5′ capped transcripts [%] among all mRNAs tested for each particular gene are shown in red. In total, 306 individual cDNAs were used for this analysis (Supplementary Figs. S1–S4; Supplementary Tables S2, S4). Chemically synthesized control RNA oligonucleotides bearing 17 adenosine residues at the 5′ end and containing a triphosphate (ppp) or cap (m7G) structure at the 5′ end are shown in blue.

subsequent purification steps and storage. In our previous experiments, we did not see any uncapped transcripts in cellular mRNAs represented by actin mRNA control[33]. Similarly, we did not detect any uncapped transcripts from VACV early genes without the INR region (Fig. 2, Supplementary Fig. S1 and Table S2). Altogether, these results confirmed the reliability and precision of the used 5′ RACE method.

As another control experiment, we wanted to check if VACV mRNAs from different GTCs contain blocked 5′ ends or if their 5′ ends

contain only natural phosphates derived from the first inserted ribonucleoside triphosphates. We took equal amounts of total RNA preparation from VACV-infected HeLa cells and subjected them either to treatment by RNA 5′ Polyphosphatase (Epicenter Biotechnologies) only or by RNA 5′ Polyphosphatase with subsequent treatment by Terminator 5′-Phosphate-Dependent Exonuclease (Epicenter Biotechnologies). Figure 5 demonstrates semiquantitative RT-PCR detection of transcripts belonging to different GTCs upon RNA 5′ Polyphosphatase and Terminator treatment (Fig. 5). While the detected amount of *J6R* (early gene) mRNA remains the same after RNA 5′ Polyphosphatase and combined RNA 5′ Polyphosphatase and Terminator treatments, *A2L* mRNA (intermediate gene) decreases, and *C3L* mRNA disappears after the combined treatment with both enzymes. This result can be interpreted that while *J6R* mRNAs and part of *A2L* mRNAs have blocked 5′ ends e.g., by m7G caps, *C3L* mRNAs possess unblocked 5′ tri-, di- or monophosphate ends.

The observed negative correlation between the VACV 5′ mRNA poly(A) leader length and 5′ cap occurrence led us to hypothesize that extended 5′ polyadenylation of VACV mRNA directly affects m7G cap synthesis. To check this hypothesis, we decided to compare the average length of capped and uncapped late VACV mRNAs, among which we expected the most striking difference in the lengths of capped and uncapped 5′ poly(A) leaders. The low occurrence of cap moieties at the 5′ ends of VACV late transcripts (Figs. 1C, 2, 5, Supplementary Table S2) makes the determination of the 5′ leader lengths of their capped variants technically challenging because it would require sequencing of unrealistically large numbers of 5′-RACE clones. Therefore, we decided to enrich 5′ capped transcripts of the selected VACV late genes by the RNA ligase-mediated amplification of cDNA ends (RLM-RACE) technique. RLM-RACE (*a.k.a.* oligo-capping) enables amplification of minute amounts of cDNA derived only from capped mRNAs by employment of three specific subsequent enzymatic reactions preceding the reverse transcription step. This procedure leads to a replacement of the original 5′ mRNA cap moiety with a short RNA oligonucleotide, which is then used after reverse transcription for subsequent PCR amplification[38]. Identical samples of total RNA purified from VACV-infected and mock-infected HeLa cells were used for the 5′-RACE and RLM-RACE techniques. Combined analyses of the 5′ RACE and RLM-RACE results clearly demonstrated that 5′ capped transcripts of both selected late VACV genes (*A17L* and *C3L*) tended to contain shorter 5′

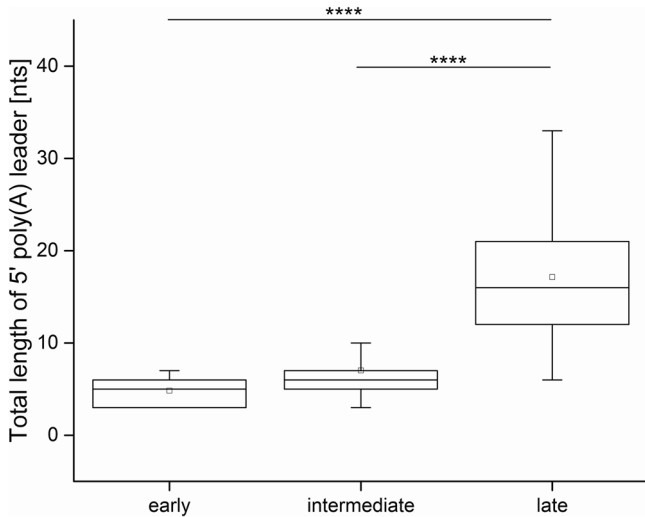

**Fig. 3 | Increasing length of the 5′ poly(A) leaders in mRNAs of each successive VACV GTC.** The box-whisker plot represents the total length (both templated and nontemplated adenosines) of 5′ poly(A) leaders in transcripts of VACV early, intermediate and late genes. The lower bar, upper bar, bottom and top of the boxes represent the 10th percentile, 90th percentile and the first and third quartiles, respectively. The square dot and inner line in each box represent the mean and median, respectively. **** indicates $p_{adj} < 0.0001$. In total, 219 cDNAs corresponding to gene promoters containing the initiator region were used for this analysis (Supplementary Tables S3, S9). A nonparametric Kruskal-Wallis test, followed by the Dunn post hoc test with Benjamini-Hochberg FDR $p$-value adjustment, revealed statistically significant differences in the 5′ poly(A) leader length between late genes and early genes ($p_{adj} < 0.0001$), and late genes and intermediate genes ($p_{adj} < 0.0001$). The difference in 5′ poly(A) leader length between early and intermediate genes was not statistically significant but approached significance ($p_{adj} = 0.0522$).

poly(A) leaders than transcripts that lacked a 5′ mRNA cap moiety; this difference was statistically significant for the *C3L* mRNA at the $p < 0.01$ level (Fig. 6 and Supplementary Table S6).

To better understand the mechanism of 5′ poly(A) leader synthesis and its potential competition with VACV mRNA capping, we investigated the possible role of the adenosine-rich INR in these processes. The INR oligo(A) stretch is commonly considered to be responsible for VACV RNAP sliding and thus for 5′ poly(A) leader synthesis. We wanted to test whether disruption of the continuous INR oligo(A) element can affect the length of the 5′ poly(A) leaders or the frequency of capping of VACV postreplicative mRNAs. We constructed reporter vectors bearing a gene encoding enhanced green fluorescent protein (EGFP), the expression of which was controlled by the viral *G8R* intermediate promoter containing either its wild-type INR sequence (TAAAAT; *pG8R^P^*-EGFP vector) or the point-mutated INR (TAA̲CAT; *pG8R^PM^*-EGFP vector). We transfected these reporter vectors into VACV-infected HeLa cells, purified total RNA 12 hours post-infection and subjected this RNA preparation to 5′ RACE analysis as described above. At least 19 independent cDNA clones corresponding to each wild-type and mutated INR variant were analyzed (Fig. 7A). Consistent with our expectation, the median lengths of the 5′ poly(A) leaders were comparable among mRNAs, the transcription of which was directed either from the viral genomic *G8R* promoter (3 nt) or its *G8R^P^* vector counterpart (2.5 nt) (Fig. 7B and Supplementary Table S7). In addition, the proportion of 5′ nontemplated poly(A) leader-containing mRNAs was comparable in both these groups: 88.9% of viral *G8R* transcripts and 83.3% of *G8R^P^* vector mRNAs were 5′ oligoadenylated (Fig. 7C and Supplementary Table S7). In contrast, a single point mutation in the center of *G8R* INR (*G8R^PM^*, TAA̲CAT) led to a significant decrease in the median and mean of the number of nontemplated adenosine

nucleotides added to the 5′ ends of the reporter mRNAs (0 and 0.3, respectively) (Fig. 7B and Supplementary Table S7) and to a significant decrease in the fraction of 5′ polyadenylated mRNAs (Fig. 7A). The ratio of 5′ polyadenylated mRNAs to all *pG8R^PM^*-EGFP transcripts dropped to 31.6% (Fig. 7C). Consistent with the observed negative correlation between the length of 5′ poly(A) leaders and the occurrence of 5′ capping, reporter mRNAs controlled by point-mutated *G8R^PM^* INR were more frequently 5′ capped than reporter mRNAs, the transcription of which was directed by the wild-type promoter *G8R^P^* (57.9% vs 20.8%, respectively) (Figs. 7A, 7D and Supplementary Table S7).

The mutational experiment clearly suggested an important role of the intact INR containing a slippery adenosine stretch for both the 5′ poly(A) leader formation and cap absence at VACV mRNAs, and thus a possible role of VACV RNAP reiterative transcription initiation in both these processes. Nevertheless, the contribution of other viral and host factors to the synthesis of uncapped VACV mRNAs cannot be excluded. VACV codes for two decapping enzymes, D9 and D10, which have been proposed to play a role in viral virulence and selective translation of viral mRNAs[39–42]. To test the hypothetical role of D9 and D10 decapping enzymes in the occurrence of uncapped and 5′ polyadenylated VACV mRNAs, we performed 5′ RACE analysis of VACV mRNAs isolated from BHK-21 cells infected with vD9muD10mu double VACV mutant. The vD9muD10mu virus contains inactivating mutations in the catalytic sites of both D9 and D10 decapping enzymes and exhibits replication defects in human and monkey cell lines, which are normally used for the VACV propagation[43]. Comparison of 5′ RACE results obtained from a cell infected with wild-type WR virus and vD9muD10mu double-mutant virus doesn't show significant differences in 5′ poly(A) leader length and m7G cap occurrence in transcripts belonging to all three GTCs (Figs. 4, 8 and Supplementary Table S2). Differences in 5′ cap occurrence and poly(A) leader length in *J6R* and *A2L* transcripts arose from preferential utilization of different TSS in wild type WR strain and vD9muD10mu double mutant (Supplementary Figs. S1, S2).

According to our results, we can conclude that a substantial portion of intermediate and the majority of late VACV transcripts are uncapped. This observation remains unaffected by the presence or absence of functional VACV decapping enzymes D9 and D10. Length of the 5′ poly(A) leaders negatively correlates with the m7G cap occurrence. Synthesis of long 5′ poly(A) leaders and uncapped VACV mRNAs are collectively dependent on the presence of the intact oligo(A) stretch within the INR site, which suggests a possible role of VACV RNAP reiteration during transcription initiation.

## Discussion

VACV was among the first DNA viruses in which 5′ capped viral mRNAs were discovered almost four decades ago, and since then, there has been a common belief that all VACV transcripts of early and postreplicative genes are capped at their 5′ ends. These findings inspired numerous fundamental investigations and discoveries on 5′ mRNA cap structures and capping machineries in other viruses and eukaryotic organisms. Our work revises this paradigm and suggests a more accurate view of the transcription-coupled regulation of viral mRNA processing within different GTCs. We demonstrate here that 5′ cap occurrence in viral mRNAs gradually decreases in each successive GTC, in contrast to the reciprocal increase in 5′ poly(A) leader lengths, and that these two variables are mutually negatively correlated. We also demonstrate that the INR element directly or indirectly influences both the frequency of 5′ mRNA capping and the occurrence of 5′ poly(A) leaders, including their lengths in postreplicative VACV mRNAs.

These findings are consistent with experiments showing a low requirement of VACV late mRNAs for the functional cap-dependent translation initiation pathway[28,44] and their dependence on virus-induced phosphorylation of receptor for activated C kinase 1 (RACK1) and ribosomes reprogrammed by virus infection[29–31]. Correspondingly,

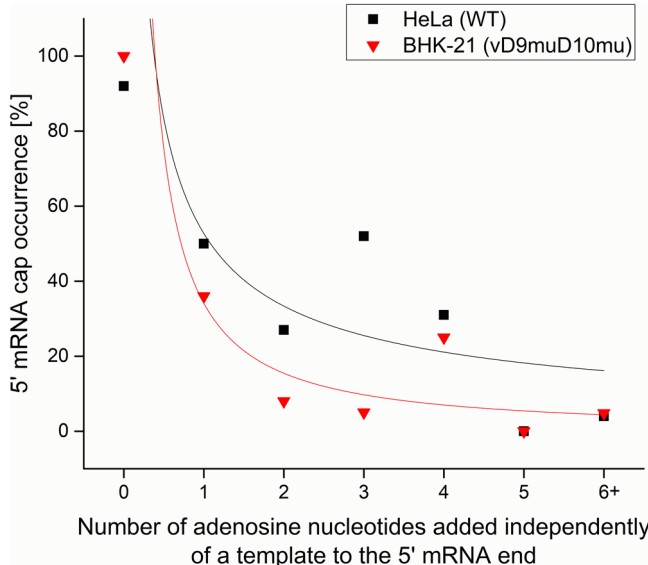

**Fig. 4 | Negative correlation between the number of nontemplated adenosines in the 5′ poly(A) leader and the presence of a 5′ mRNA cap at the end of VACV mRNAs.** The plot shows the proportion of 5′ mRNA cap structures occurring in the transcripts of all GTCs. 6 + corresponds to 6 and more nontemplated adenosines in the 5′ mRNA leader. In total, 306 viral mRNA sequences from HeLa cells infected with WR VACV (WT; black) and 185 viral mRNA sequences from BHK-21 cells infected with vD9muD10mu double mutant (red) were used for this analysis (Supplementary Table S4).

VLE mRNAs containing 5′ poly(A) leaders are loaded on polysomes independently of the cap-binding eIF4E translation initiation factor[33]. Recently, Cantu et al. showed that inactivation of vaccinia virus decapping enzymes D9 and D10 results in decreased loading of intermediate and late vaccinia virus transcripts to polysomes and decreased cellular levels of core protein 4a and virion membrane protein A17, which are encoded by vaccinia virus late genes *A10L* and *A17L*, respectively[41]. This experiment can be, besides other effects on the cellular antiviral machinery, interpreted as a situation when cellular cap-dependent translation is strengthened. Its result is consistent with inverse experiments showing that inhibition of cap-dependent translation conversely leads to increased translation of vaccinia virus late transcripts and VLE mRNAs containing 5′ poly(A) leaders[28,33]. The easiest interpretation of these experiments is that poxviral postreplicative mRNAs are, similarly to their VLE relatives, 5′ polyadenylated and uncapped[32,33].

This interpretation is further supported by our results showing that 5′ leaders of the infrequently occurring capped late VACV transcripts (Figs. 2, 5 and 8) were markedly shorter than 5′ leaders of the dominant uncapped transcripts derived from the same genes (Fig. 6). These results might indicate the possibility that the length of the 5′ poly(A) mRNA leaders of VACV postreplicative transcripts calculated from whole-genome analyses of the VACV transcriptome might be underestimated, because only mRNAs containing the 5′ cap were used for deep sequencing[37]. The medians of VACV intermediate transcript 5′ poly(A) leader lengths determined by Yang et al. and us are comparable and span 8 nt and 7 nt, respectively[37] (Supplementary Fig. S2). However, the medians of VACV late transcript 5′ poly(A) leader lengths determined by cap-independent 5′ RACE were between 20 nt and, as such, were significantly longer than those determined by the cap-selective deep-sequencing method (11 nt, Supplementary Fig. S3)[37]. The latter readout is also in good agreement with the results obtained by us using the cap-selective RLM-RACE approach, collectively yielding a median length of 8 nt for the VACV A17L and C3L late mRNA 5′

poly(A) leaders (Fig. 6, Supplementary Fig. S5 and Supplementary Table S6).

We suggest that vRNAP slippage at the INR site and thus synthesis of poly(A) leaders at 5′ ends of the intermediate and late poxviral mRNAs can directly affect the synthesis of the 5′ cap because nontemplated poly(A) leader does not pair well with the DNA template strand in the vRNAP active site and exits vRNAP through the non-canonical path which prevents its capping by VACV capping enzyme sitting close to the canonical exit channel[15]. This phenomenon has been well studied, including structural models as reiterative transcription initiation in transcribing bacterial RNAPs[45,46]. Consistent with our observations, we hypothesize that reiterative transcription initiation by VACV RNAP at late promoters does not preclude the occasional capping of 5′ polyadenylated mRNA. This could occur in various ways, including viral mRNA randomly entering a canonical RNAP exit channel or reaching the capping enzyme active site via a different route outside the RNAP. The idea that the degree of 5′ mRNA polyadenylation inversely influences synthesis of the 5′ mRNA cap is further supported by our observation that 5′ poly(A) leaders in m7G cap-containing VACV late transcripts were shorter than the 5′ mRNA leaders, the lengths of which were calculated from the unbiased set of all VACV late mRNAs. Collectively, our results support the hypothesis that VACV transcription regulation ensures a gradual shift in viral mRNA translation initiation from a cap-dependent to cap-independent mechanism[28,44], which is accompanied by virus-induced modification of the host translation machinery[29–31]. This hypothesis is further supported by recent observations that inactivation of the vaccinia virus decapping enzymes D9 and D10 affects translation of intermediate and late mRNAs but has no effect on translation of early viral mRNAs[41]. Many viruses try to overcome host protein synthesis shutoff induced by viral infection and usurp the host resources and molecular machineries for preferential synthesis of viral proteins. Poxviruses are not an exception[47]. It might be possible that the function of the vaccinia virus decapping enzymes is to lower the proportion of capped cellular mRNAs competing with viral uncapped transcripts for the cellular translation apparatus, and at the same time to produce decoy uncapped mRNAs for the mRNA degradation pathways, which may increase the lifetime of uncapped viral postreplicative mRNAs. Vaccinia virus 3′-poly(A)-polymerase promiscuously uses various viral and cellular RNAs to generate small nontranslated polyadenylylated RNAs (POLADS), which are supposed to inhibit host protein synthesis by sequestering poly(A)-binding protein (PABP)[47,48]. Interestingly, poxvirus-related VLE's 5′-polyadenylated and uncapped mRNAs can be translated independently of the cap-binding eIF4E translation initiation factor, and production of VLE proteins encoded by uncapped and 5′ polyadenylated mRNAs increases in cells lacking PABP and Lsm1 proteins[33]. This may explain why sequestering PABP by POLADS and decoying cellular mRNA degradation pathways can provide additional advantage to poxviral mRNAs in their competition for the cellular translation machinery. Although published data clearly suggest that vaccinia virus uncapped and 5′ polyadenylated mRNAs could be loaded onto polysomes and translated by an eIF4E-independent pathway, these experiments have yet to be performed. We are currently working in this direction.

Overall, the available data highlight the importance of the INR region in the transcription of intermediate and late VACV genes. Baldick et al. and others have shown that mutations in the INR region are essential for vaccinia promoter function and gene expression[17,36]. We point-mutated the TAAAAT motif of the G8R INR by substituting A with C at the + 3 position (numbered according to Baldick et al., 1992[17]). Thanks to the RT-PCR amplification step included in the 5′ RACE method, we were able to detect specific transcripts driven by the mutated G8R promoter and successfully analyze their 5′ ends. Analysis

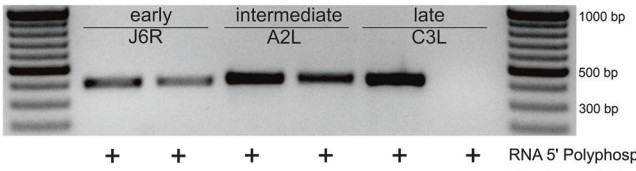

**Fig. 5 | Analysis of the 5′ ends of VACV mRNAs.** RNA purified from HeLa cells infected with the VACV WR strain was treated either by RNA 5′ Polyphosphatase or by RNA 5′ Polyphosphatase followed by Terminator 5′-Phosphate-Dependent Exonuclease and analyzed by semiquantitative RT-PCR (21 amplification cycles) for the presence of early (J6R), intermediate (A2L) and late (C3L) transcripts. The figure shows the results of one of two independent experiments.

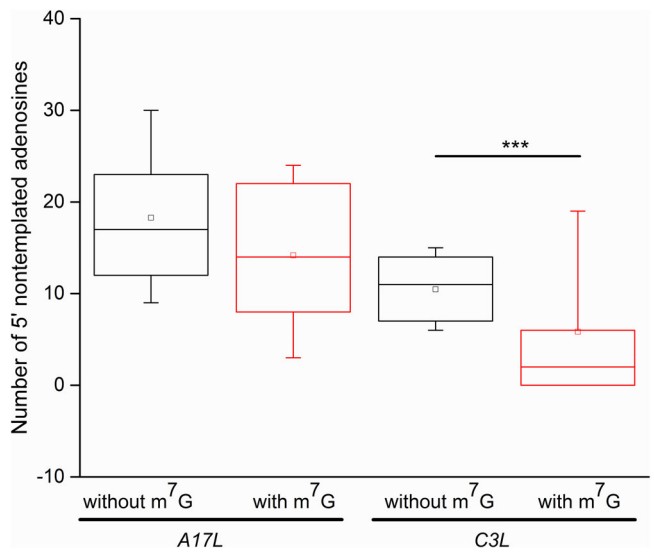

**Fig. 6 | 5′ capped late VACV mRNAs contain shorter 5′ poly(A) leaders than their uncapped counterparts.** The box-whisker plot represents the lengths of non-templated 5′ poly(A) leaders in *A17L* and *C3L* mRNAs that either contained (in red) or did not contain (in black) the 5′ cap moiety ($m^7G$); the general description of the box plot is the same as that in Fig. 2. In total, 96 sequences were used for the analysis (Supplementary Fig. S5 and Supplementary Table S6). A nonparametric Mann-Whitney U test revealed statistically significant differences in the 5′ poly(A) leader length between $m^7G$-capped and uncapped *C3L* mRNAs ($p = 0.00096$; ***).

revealed that changing the G8R INR motif from TAAAAT to TAACAT caused the disappearance of 5′ poly(A) leaders and increased mRNA capping (Fig. 7). We also constructed a reporter bearing a double-mutated TACCAT motif within the G8R promoter; however, this promoter was not functional, and we could not detect any specific transcripts, even with nested RT-PCR amplification. Our results confirm the importance of an intact INR region for the transcription of post-replicative genes and suggest a possible important role for 5′ poly-adenylated and uncapped mRNA in VACV gene expression during later stages of infection. Consistent with these results, we previously demonstrated that mutating the TAAAAT INR motif sequentially to TAACAT and TACCAT within the K1UCR2 promoter of pGKL VLE led to decreased 5′ mRNA polyadenylation and increased 5′ mRNA capping. Unlike VACV, these mutations were viable and did not prevent transcription and gene expression from the pGKL VLE promoter[32,33]. Interestingly, Ink et al. showed that shortening the INR motif of the 38K cowpox promoter inserted into the recombinant VACV abolished the formation of a 5′ poly(A) leader, but not promoter function[49].

## Methods
### Cells
Epithelial cell lines HeLa (human cervical carcinoma) and BSC-40 (African green monkey kidney cells) were grown in Dulbecco's

modified Eagle's medium (DMEM; glucose 4.5 g/l supplemented with 10% heat-inactivated fetal bovine serum (10% FBS-DMEM) or neonatal calf serum (10% NCS-DMEM), respectively. BHK-21 fibroblasts (Baby hamster kidney-21 cells) were grown either in Minimum essential medium with Earle's balanced salts (EMEM) supplemented with 2 mM L-glutamine and 10% FBS (10% FBS-EMEM) or in 10% FBS-DMEM. The media included penicillin ($1 \times 10^5$ U/l) and streptomycin (100 mg/l). The cells were maintained at 37 °C, in a 5% $CO_2$ atmosphere with 95% humidity. All the media and growth supplements were purchased from Gibco BRL or Sigma-Aldrich.

### Viruses, infection and transfection
A purified stock of wild-type VACV strain Western Reserve (WR) was used throughout the study. Additionally, WR mutant vD9muD10mu with catalytic site mutations in both VACV decapping enzymes, which was kindly provided by Bernard Moss[43], was used. The wild-type WR was propagated in BSC-40 cells supplemented with 2% NCS-DMEM as described previously[50,51], purified by sucrose gradient sedimentation[52] and the titer of the virus was determined by serial dilution and plaque assays in BSC-40 cells. WR mutant vD9muD10mu was grown and titrated using plaque assay in BHK-21 cells supplemented with 2.5% FBS-EMEM[43].

For experiments, $1.5 \times 10^6$ HeLa and BHK-21 cells were seeded in 60 mm plates a day before infection. HeLa cells were either mock-infected or infected with WR at a multiplicity of infection (m.o.i.) 5 for 40 min, washed with DMEM and supplemented with 10% FBS-DMEM[53]. When appropriate, cells were transiently transfected with the respective plasmids at 5-10 μg/plate using polyethylenimine according to ref. 54. HeLa and BHK-21 cells were mock-infected or infected with vD9muD10mu at m.o.i. 10 for 45 min, washed with DMEM and supplemented with 2% FBS-DMEM. The culture medium was removed, and the cells were lysed in RNA Blue (Exbio) at the indicated times after infection (4 or 12 h post-infection).

### RNA purification, reverse transcription and 5′ RACE
Total RNA from RNA blue lysis solution was purified according to the manufacturer's protocol. DNA was removed using a DNA-free Kit (Ambion) according to the manufacturer's protocol. The integrity of the total RNA was analyzed using agarose electrophoresis as described in ref. 55. 5′ RACE-PCR experiments were performed exactly according to ref. 33. Briefly, SuperScript III reverse transcriptase, which belongs to a group of reverse transcriptases that are able to overcome the 5′–5′ bond between the mRNA cap structure and the very first mRNA nucleotide[56], was used for cDNA synthesis from total RNA using random hexamer primers (Invitrogen). After reverse transcription, the cDNA was purified using a High Pure PCR Product Purification Kit (Roche) and used for subsequent 5′ RACE-PCR amplification with gene-specific primers (Supplementary Table S1). Oligo-capping experiments were performed exactly according to ref. 33. The PCR amplicons were separated by agarose gel electrophoresis; each specific product was purified from the gel, inserted into the pCR4-TOPO vector (Invitrogen) and sequenced. At least 15 independent clones for individual VACV

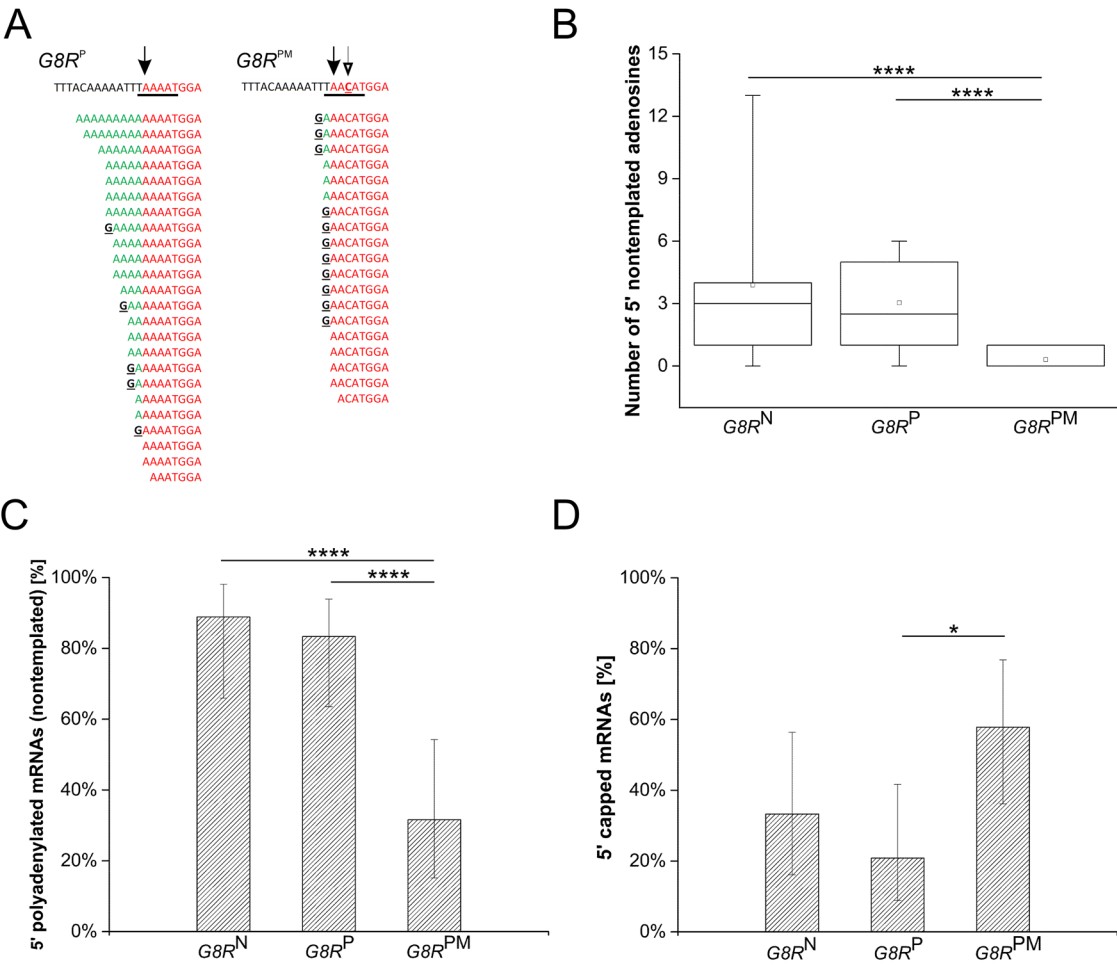

**Fig. 7 | VACV INR controls 5′ end formation of viral mRNAs. A** 5′ RACE analysis of EGFP reporter mRNAs transcribed from the intermediate wild-type *G8R* promoter (*G8R^P*) or its point-mutated version (*G8R^PM*). Single-nucleotide A/C substitution in *G8R* INR (empty arrow, underlined in red) resulted in the reduction in the number of added nontemplated adenosine residues and dramatic shortening of the 5′ poly(A) leaders. The upper sequence corresponds to the viral template DNA. TSS and INR are marked with a filled black arrow and a black underline, respectively. The sequences below represent individual sequenced cDNA clones. The 5′ untranslated regions are shown up to the ATG translation start codon. Nucleotides identical to the viral template DNA are labeled in red. Nucleotides added in a nontemplated manner are labeled in green. The guanosine residues corresponding to the 5′ mRNA cap are marked in black. **B** Statistical representation of panel (**A**), including results from 5′ RACE analysis of natural viral *G8R* mRNAs (*G8R^N*; Supplementary Fig. S2 and Supplementary Table S2) for comparison. The general description of the box plot is the same as that in Fig. 2. **** indicates $p_{adj} < 0.0001$. A nonparametric Kruskal-Wallis test, followed by the Dunn post hoc test with

Benjamini-Hochberg FDR *p*-value adjustment revealed statistically significant differences in the 5′ poly(A) leader length between *G8R^PM* and *G8R^N* mRNAs ($p_{adj} = 0.000037$) and *G8R^PM* *G8R^P* mRNAs ($p_{adj} = 0.000037$). **C** Single A/C substitution in *G8R* INR leads to a significant reduction in the 5′ poly(A) leader occurrence and its length. Bars represent the percentages of 5′ polyadenylated transcripts among all mRNAs transcribed from the *G8R^N*, *G8R^P*, and *G8R^PM* promoters. Error bars depict 95% confidence intervals; **** indicates $p \leq 0.0001$. **D** Single A/C substitution in *G8R* INR increases the occurrence of 5′ capped transcripts. Bars represent the percentages of 5′ capped mRNAs transcribed from the *G8R*, *G8R^P*, and *G8R^PM* promoters. Error bars depict 95% confidence intervals; * indicates $p < 0.05$. In total, 61 sequences were used for the analyses depicted in panels (**B**–**D**) (Supplementary Tables S7, S8). Two-sided Fisher's exact test showed statistically significant association of point-mutated INR with low occurrence of poly(A) leaders (*G8R^PM* x *G8R^N*, $p = 0.0001$; *G8R^PM* x *G8R^P*, $p = 0.0001$) and higher occurrence of m7G cap (*G8R^PM* x *G8R^P*, $p = 0.0248$) in *G8R^PM* mRNA.

gene transcripts were analyzed. All primers used in this study are listed in Supplementary Table S1.

### Chemically synthesized control RNA oligonucleotides

Chemically synthesized control RNA oligonucleotides, containing either a 7-methylguanosine cap (m7G-RNA) or a triphosphate (ppp-RNA) at the 5′ end, were synthesized by Bio-Synthesis Inc, USA as a custom RNA oligonucleotides (for sequence see Supplementary Table S1). For analysis, 0.25 µg of each oligonucleotide was used for reverse transcription using SuperScript III reverse transcriptase and RT_for_RNA_oligo primer. After reverse transcription, the cDNA was immediately used for subsequent 5′ RACE-PCR amplification using RNA_oligo_nested_I primer as described above.

### 5′ end RNA analyses

Total VACV RNA (12 h.p.i) from RNA blue lysis solution was purified according to the manufacturer's protocol. Approximately 3.5 µg of DNAse-treated RNA (as above) was incubated with 40 U of RNA 5′ Polyphosphatase (Epicenter Biotechnologies) for 30 minutes at 37 °C, purified using acidic phenol/chloroform extraction and precipitated using ammonium acetate and ethanol. Precipitated RNA was washed with 70% ethanol, resuspended in 30 µl of DEPC-treated ddH2O. Ten microliters of resuspended RNA was used for RT-PCR (as above); 20 µl was incubated with 2 U of Terminator 5′-Phosphate-Dependent Exonuclease (Epicenter Biotechnologies) for 60 min at 30 °C, purified using acidic phenol/chloroform extraction, precipitated using ammonium acetate and ethanol, washed with 70% ethanol,

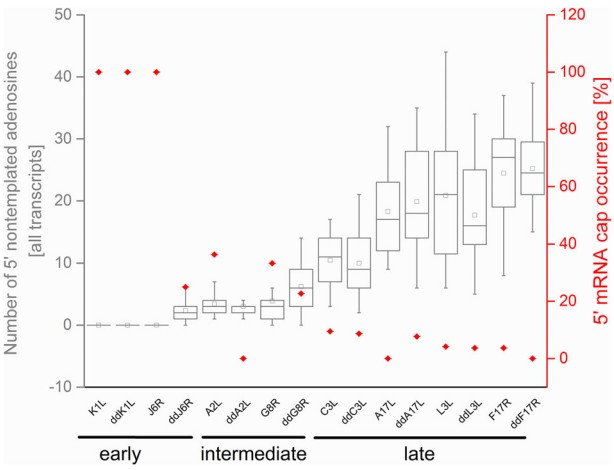

**Fig. 8 | A side-by-side comparison of the 5′ poly(A) leader length and the occurrence of the m⁷G cap in the VACV transcripts of the wild-type and vD9muD10mu strains.** The plot shows a comparison of transcripts purified from HeLa cells infected with the VACV WR strain and transcripts purified from BHK-21 cells infected with the vD9muD10mu double mutant (marked as *dd*). Similarly as in Fig. 2, data from both VACV strains show a negative correlation between the 5′ poly(A) leader length (left Y-axis) and cap occurrence (right Y-axis) in mRNAs from different successive GTCs represented by selected early, intermediate and late vaccinia virus genes (X-axis). The box-whisker plot (in gray) represents the lengths of nontemplated 5′ poly(A) leaders. The lower bar, upper bar, bottom and top of the boxes represent the 10th percentile, 90th percentile and the first and third quartiles, respectively. The square dot and inner line in each box represent the mean and median, respectively. The proportions of 5′ capped transcripts [%] among all mRNAs tested for each gene are shown in red. In total, 359 individual cDNAs were used for this analysis (Supplementary Figs. S1–S3 and Supplementary Table S2).

resuspended in 10 μl of DEPC-treated ddH₂O and used for RT-PCR (as above). For the following PCR amplification (3 min at 95 °C; 21 cycles of 30 s at 94 °C; 30 s at 50 °C; 60 s at 72 °C; and finally, 10 min at 72 °C), 0.33 μl of the reaction mixture was used with an appropriate gene-specific primer pair. Amplification products (5 μl) were analyzed using 1.5% agarose gel electrophoresis.

### Reporter plasmid construction
The promoter-less plasmid pEGFP-N1(-P)[57] was digested with the BamHI and SalI restriction endonucleases to insert either the VACV G8R intermediate promoter sequence (G8RP; using VV_G8Rp_1/VV_G8Rp_2 oligonucleotides) or the G8RMP promoter bearing a single point mutation within the INR (oligonucleotides VV_G8Rp_mut_1/VV_G8Rp_mut_2). Plasmids were verified by sequencing.

### Statistical analyses
Statistical analyses were performed exactly according to ref. 33. Briefly, the variance in the 5′ poly(A) leader lengths of selected VACV mRNAs was analyzed using the nonparametric Mann-Whitney U test. For multiple comparisons, the nonparametric Kruskal-Wallis test was used, followed by the post hoc Dunn test with *p*-value adjustment according to the Benjamini-Hochberg FDR method. These data did not follow a normal distribution according to the Shapiro-Wilk test (Figs. 3, 6; 7B). Categorical binary data of 5′ mRNA cap and 5′ poly(A) occurrence in VACV transcripts were evaluated using two-tailed Fisher's exact test with the 95% confidence interval calculated using the adjusted Wald method (Fig. 7C, D).

### Reporting summary
Further information on research design is available in the Nature Portfolio Reporting Summary linked to this article.

## Data availability
All data, including the sequencing data, can be found in the manuscript and supplementary files. The figures and Supplementary Figures contain all the sequences with highlighted nontemplated adenosines and m⁷G caps that were used for the analyses. The data are numerically abstracted in Vopalensky_2025_data_source_file.xlsx. Source data are provided in this paper.

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

## Acknowledgements

We thank Petra Studnickova, Natalie Suchankova, Monika Kaplanova, and Libor Krasny for their invaluable help, and Bernard Moss for generously providing mutant viruses. Funding was provided by the Czech Science Foundation (grant No. 21-25504S to V.V.); the ELIXIR CZ research infrastructure project (MEYS grant no. LM2023055, to M.P.), including access to computing and storage facilities; and the project National Institute of Virology and Bacteriology (Program EXCELES, ID Project No. LX22NPO5103)—funded by the European Union—Next Generation EU to M.P.

## Author contributions

M.S., V.V., and M.P. devised the project, main conceptual ideas and methods of data analysis; M.S., Z.M., I.B., K.H., T.M., and V.V. performed experiments; V.V., M.S., and M.P. analyzed data and created the figures;

and V.V. and M.P. wrote the original draft. All authors discussed the results and edited the manuscript.

## Competing interests

The authors declare no competing interests.
