## [Transparent Peer Review file · Nature Communications]

Vaccinia virus mRNAs containing 5' polyadenosine leaders lack a methylguanosine cap

Corresponding Author: Dr Martin Pospisek

Version 1:

Reviewer comments:

Reviewer #1

(Remarks to the Author)

Several previous studies showed, by a variety of direct and indirect methods, that vaccinia virus early and post-replicative mRNAs are capped. The claim of the authors, as stated in the title, that transcripts of vaccinia virus post-replicative genes do not contain a 5' methylguanosine cap, is surprising and therefore requires rigorous supporting data. Unfortunately, nearly all their data depend on an experimental method that lacks necessary controls and validation.

Comments

1. The main conclusion of the paper is that capping is inversely proportional to length of the 5' poly(A) tract. However, an alternate interpretation of the data is that the detection of caps is inversely proportional to the length of the 5' poly(A). For most of the experiments, the comparison of capped and uncapped mRNA depends on an empirical method for enriching full-length mRNAs described by Schmidt and Mueller (1999) called CapSelect and is referred to as 5' RACE in this paper. The method critically depends on the ability of reverse transcriptase in the presence of Mn⁺⁺ to jump over the 5' to 5' triphosphate and add 3 to 5 C residues to the m7G of the majority of capped RNAs. This is followed by ribonucleotide tailing with terminal transferase, ligation of an adaptor, PCR amplification and sequencing. In the Schmidt and Muller paper, the method was verified with a few human mRNAs. My concern is that the efficiency of C-addition was not tested for RNAs in which the cap is preceded by a long run of adenylates. Reverse transcriptase can have difficulties with long homopolymeric stretches and if pausing or dissociation occurs before the cap, then it will be scored as uncapped. In fact, Scher and Stunnenberg (1987) reported that the length of the 5' poly(A) determined using reverse transcriptase was much shorter than that found using other methods. In brief, I suggest that the efficiency of 5' RACE is inversely proportional to the length of poly(A), thereby giving the illusion of a deficiency in capping. The authors need to verify the 5' RACE procedure with RNAs in which the cap is directly preceded by runs of adenylates of varying length. This control was also not done for their earlier papers on yeast VLEs.

2. The authors never discuss the structure of the putative uncapped RNAs. Presumably, the RNAs have 5' triphosphates. This should be demonstrated and there is a simple way to do this. RNAs with 5' tri- or diphosphates can be capped in vitro with the vaccinia virus capping enzyme, which is available commercially. Indeed, diphosphate ended poly(A) was used as the acceptor substrate in the original paper on purification of capping enzyme (Martin et al. 1975). If the model of the authors is correct, in vitro capping of the RNAs should increase the percentage of capped RNAs detected by the 5' RACE method. If, on the other hand, the RNAs are fully capped, then removal of the caps by periodate oxidation and beta elimination or enzymatically would leave a diphosphate end that could be capped in vitro as shown by Ahn and Moss (1989).

3. A second method, oligo capping (RLM-RACE), was also used to analyze the RNAs of the post-replicative C3L and A17L genes. The lengths of the poly(A) determined by RLM-RACE were compared to that produced by RACE. In the case of A17L, the length of the poly(A) determined for the capped RNA by RLM-RACE was not statistically different from the total RNA determined by RACE. This contradicts their finding that little or no capped A17L RNAs were detected by the 5' RACE method. A statistical difference of $p < 0.01$ was found for the C7L RNA.

4. The model that purports to explain decreased capping on pages 9 and 10 and figures 5 and 6 is built on an erroneous supposition that intermediate and late genes are transcribed on an RNA polymerase complex that contains the early transcription factor (VETF) and associated RAP94. There are no data suggesting that intermediate and late transcription

occur on an RNA polymerase complex that contains either VETF or RAP94. In fact, VETF and RAP94 are expressed following DNA replication and intermediate and late transcription can occur under conditions in which DNA replication is prevented (Keck et al. 1990). Furthermore, intermediate and late transcription factors as well as the capping enzyme and cap methyltransferase are associated with the vaccinia virus replisome isolated by iPOND, but neither VETF nor RAP94 are present (Senkevich et al. 2017). Belatedly, the authors acknowledge that a different RNA polymerase complex may be used for intermediate and late transcription. Pages 9 and 10 and figures 5 and 6 and subsequent discussion should all be deleted as they will only confuse a reader.

5. p. 5 Lines 6 - 7. Statement that “the presence of the 5' cap structure in individual VACV transcripts has never been tested” is incorrect.

A. Ahn and Moss (1989) analyzed total late mRNA and specifically the late F18 (11 kDa) mRNA. They used a decapping-radioactive recapping method and provided evidence that F18 and all late mRNAs are capped supporting the original report of Boone and Moss (1977).

B. Scher and Stunnenberg (1987) proved that the late 11K (F18L) and 4b (A3L) mRNAs with poly(A) leaders have caps by using anti-cap antibody to immunoprecipitate cDNA-RNA hybrids. In that paper they also showed a discrepancy in the determined length of the poly(A) when reverse transcriptase was used in which it appeared to consist of only 5 to 10 adenylates whereas the length was about 35 adenylates using other methods. The difference was attributed to inefficiency of reverse transcriptase to copy a poly(A) template.

C. Yang et al (2011) showed by CAGE methods that early mRNAs, including rare ones with 5' poly(A), have caps. In another paper Yang et al (2012) identified caps on ~100 individual intermediate and late mRNAs followed by poly(A) by CAGE. These data included the four genes analyzed in the present study: G8R, A1L, C3L and A17L.

6. The title is misleading - based on their data it should be “Transcripts of vaccinia virus postreplicative genes may lack a 5' methylguanosine cap”. It should be noted that each of the four intermediate and late genes analyzed in this study were shown to have caps either by the 5' RACE or RLM-RACE method.

7. p.6, lines 8 - 12. The experiments of Cantu et al are more complex than indicated and would be difficult for the reader to judge. In the absence of decapping enzymes, large amounts of double-stranded RNA accumulate, and KO of RNase L and PKR only partially restores replication suggesting additional host defenses as the simplest explanation. These sentences should be deleted

8. P8, lines 20 and Fig. 4. Should point out that Baldick et al. (1992) showed that there was about ~98% inhibition of transcription when one of the A residues in the TAAAAT sequence was mutated and disrupted the AAA initiator indicating a profound effect on the transcription mechanism.

9. The Results and Discussion should be separated to better distinguish experimental observations and interpretation

Reviewer #2

(Remarks to the Author)

Manuscript NCOMMS-21-43632A-Z:

Transcripts of vaccinia virus postreplicative genes do not contain a 5'-methylguanosine cap

The authors characterized the vaccinia virus (VACV) transcripts at the individual mRNA molecule level and found that some vaccinia virus postreplicative mRNAs, containing non-templated 5' poly(A) leaders, lack the 5' cap structure in vivo. They showed for the late gene G8R that the lengths of the non-templated leaders and the presence or absence of cap structures at the 5' mRNA ends was controlled by the initiator sequence of the VACV postreplicative promoter. With molecular dynamics simulation the authors tried to support a possible link between structural features of the VACV transcribing complex and synthesis of the uncapped poly(A) mRNA leaders.

The study shows novel aspects of poxviral transcription, but should be supported by additional experiments. The authors analyzed five early genes, two intermediate and two late genes. Even within the early gene mRNAs, the 5' cap occurrence varied significantly. In addition, the intermediate and late mRNAs had different amounts of 5' cap structures. These analyses should be performed with more genes to allow for a proper statistical analysis. There are 118 early genes and 93 post replicative genes, more than 5 or 4 each should be analyzed.

The authors claim that the absence of the 5'-cap structure is due to the transcription process and support this theory by an experiment using a mutated promoter sequence of the G8R gene (G8RPM). The mutated promoter sequence resulted in more capped mRNA and less 5'-poly A. The authors should prove that the G8RPM is still transcribed at late time points and should use additional genes to support these data. The data were supported by molecular dynamics simulations using the A17L gene. Why not the G8R?

VACV is encoding two decapping enzymes; D9 expressed early and D10 expressed late. How can the authors exclude that the reduced amount of 5'-cap structures is not due to decapping, followed by RNA degradation, which would explain the lower amount of 5'-As. Experiments with D9/D10 deleted viruses or inactivated D9/D10 should be performed. In addition, cellular decapping enzymes might be active. Please consider for control experiments.

Reviewer #3

(Remarks to the Author)

Poxviruses are large cytosolic DNA viruses that encode a viral multi-subunit DNA-dependent RNA polymerase (vRNAP). Members of this family are responsible for a wide range of relevant infectious diseases in humans and animals, including monkeypox and smallpox. A thorough understanding of their unique cytosolic transcription apparatus is of general interest, not only since some of its components are widely used as a biotechnical tool (in particular the vaccinia capping enzyme). Until recently, it was a widespread belief that vaccinia transcripts are heavily modified with a cap structure at their 5' end. Here, for the first time, Vopálenký et al. use RACE PCR to examine the 5' ends of vaccinia early, intermediate and late transcripts with the surprising finding that only roughly half of the early transcripts are capped and most of the intermediate and late transcripts are not capped at all. They further provide convincing experimental evidence that the occurrence of a cap structure is strictly negatively correlated to the appearance of non-encoded poly-A stretches in the 5'-UTRs. Finally, a "slipping" mechanism is discussed that probably is an explanation for the generation of non-encoded poly-A stretches by vRNAP. This hypothesis is backed by a molecular dynamics (MD) simulation of a respective DNA/RNA hybrid in the range of the INR promoter element.

The experimental methods used in this study appear technically sound and the results are of genuine scientific interest. The manuscript therefore deserves publication after the following shortcomings have been corrected:

1. The MD simulation of the DNA/RNA hybrid results in the transformation of the double helix into an extended 'ladder'. A comparative simulation with a corresponding hybrid of a non-slipping promoter should be performed and the results presented to validate the simulation. Please describe also in the methods section, how the hybrid coordinates were generated.

2. The results of the MD simulation can only be validly discussed in the context of the vRNAP transcription machinery if the ladder conformation can be reasonably fitted in the active site cleft. A respective figure should be presented in the supplemental section.

3. The authors argue that for intermediate/late transcripts, the "Nascent 5' end of the viral mRNA thus cannot efficiently displace Rap94 B-reader element and enter the RNA exit tunnel leading to the TPase active site. This situation leads to synthesis of mRNAs with longer nontemplated 5' poly(A) 20 leaders which exit vRNAP by different way than viral early mRNAs and thus can escape the viral capping machinery" (p. 10, l. 17ff). This notion is problematic as the current understanding is that vaccinia intermediate and late transcription is not dependent on Rap94 (i. e. one would expect the exact opposite: that early transcripts frequently slip and intermediate/late do not).

Minor points:

- p. 16, l. 5: This seems to refer to the Amber ff99SB-ILDN force field.

- p. 9, l. 13: The nascent mRNA strand ...

- p. 10, l. 22: "However, we also cannot exclude a possibility" should read: "However, we also cannot exclude the possibility"

- p. 11, l. 3: "Nevertheless, only part of the vRNAPs is associated with Rap94" should read: "Nevertheless, only a part of the vRNAP population is associated with Rap94"

Version 2:

Reviewer comments:

Reviewer #1

(Remarks to the Author)

Major Comments

The manuscript provides novel and significant new data regarding the 5' structure of vaccinia virus mRNAs. In my original review, I recommended additional experiments to confirm the interpretation of the 5' RACE data by independent methods. I am now satisfied that the authors have done that, and I withdraw my major criticism.

Additional comments

1. I still recommend that the title be changed as there are two classes of post-replicative mRNAs – intermediate and late. The data presented show that 11 – 56% of various intermediate mRNAs are capped and that 0 – 10% of various late mRNAs are capped. A better title would be that transcripts with long poly(A) leaders do not contain caps or some variation of that.

2. Introduction or Results should briefly explain the basis for identifying caps using 5' RACE since many readers will not know this.

3. The experiment using 5' polyphosphatase and terminator support the 5' RACE data in showing that most late transcripts are uncapped. However, it would be informative to know whether the RNAs have a tri, di or monophosphate.

4. The authors should acknowledge that they did not analyze mRNA-associated with polyribosomes and that translation of the uncapped RNA was not directly determined in this study.

Reviewer #3

(Remarks to the Author)

In my previous review, I raised concerns regarding the inclusion and interpretation of the molecular dynamics (MD) simulations, specifically their relevance to the biological context of vaccinia virus RNA polymerase and the transcriptional processes under study.

I appreciate that the authors have carefully considered the feedback from all referees. They have now removed the entire

MD simulation section and the associated (too speculative) discussion from the manuscript. My principal concern has therefore been addressed.

In addition, the authors have expanded and strengthened their experimental data to address several points raised by the other reviewers.

While the MD simulations would, in general, have been desirable, the manuscript is of sufficient depth and quality without them.

Therefore, I am happy to recommend acceptance of the manuscript in its current form.

Reviewer #4

(Remarks to the Author)

The authors have adequately addressed the concerns laid out in the initial review.

Dear Reviewers,

We are grateful for the opportunity to work on the revision over an extended period of time. There were several milestones that we had to reach during the revision process. First, we had to obtain a new recombinant vaccinia virus and prepare everything necessary to work with it. We also followed the editor's strong recommendation to address as many of the reviewers' comments as possible in support of our unexpected and surprising results, as Reviewer 1 also stated. We substantially increased the number of VACV genes whose transcripts we subjected to 5' RACE analysis. In total, we analyzed and sequenced 491 cDNA clones derived from VACV mRNAs to obtain a complete and reliable picture of poly(A) leader and cap synthesis at the 5' ends of VACV mRNAs. We also spent a lot of time on time-consuming molecular dynamics (MD) experiments. Ultimately, however, we decided not to include the MD experiments because they were based on VACV early RNAP structures, as the VACV RNAP structures from the late stages of vaccinia infection were unavailable. Only one of three reviewers supported the MD experiments, and some of the comments challenging this part of our results were very strong. We understand the concerns and have started a project to obtain real structural data for the postreplicative VACV RNAP in the future. We also moved the remaining methods from the supplement to the main body of the manuscript and revised the introduction due to the emergence of new data. Specifically, the mpox pandemic, which we had hypothesized and predicted in the first version of the manuscript, has already occurred. Thus, the manuscript underwent substantial modifications, as is clear from the PDF with marked changes.

Answers to the reviewers' comments:

Reviewer 1

Introductory paragraph: *“Several previous studies showed, by a variety of direct and indirect methods, that vaccinia virus early and post-replicative mRNAs are capped. The claim of the authors, as stated in the title, that transcripts of vaccinia virus post-replicative genes do not contain a 5' methylguanosine cap, is surprising and therefore requires rigorous supporting data. Unfortunately, nearly all their data depend on an experimental method that lacks necessary controls and validation.”*

The 5' RACE method used throughout this work was previously published in Vopalensky et al., 2019. As I will describe below, we believe that we provided substantial supporting data for this method in the aforementioned previous publication, as well as in this manuscript. However, we took this concern seriously and conducted additional experiments, which I describe in more detail below. I believe these experiments should dispel any remaining doubts.

Comment 1: *“The main conclusion of the paper is that capping is inversely proportional to length of the 5' poly(A) tract. However, an alternate interpretation of the data is that the detection of caps is inversely proportional to the length of the 5' poly(A). For most of the experiments, the comparison*

of capped and uncapped mRNA depends on an empirical method for enriching full-length mRNAs described by Schmidt and Mueller (1999) called CapSelect and is referred to as 5' RACE in this paper. The method critically depends on the ability of reverse transcriptase in the presence of Mn²⁺ to jump over the 5' to 5' triphosphate and add 3 to 5 C residues to the m7G of the majority of capped RNAs. This is followed by ribonucleotide tailing with terminal transferase, ligation of an adaptor, PCR amplification and sequencing. In the Schmidt and Muller paper, the method was verified with a few human mRNAs. My concern is that the efficiency of C-addition was not tested for RNAs in which the cap is preceded by a long run of adenylates. Reverse transcriptase can have difficulties with long homopolymeric stretches and if pausing or dissociation occurs before the cap, then it will be scored as uncapped. In fact, Scher and Stunnenberg (1987) reported that the length of the 5' poly(A) determined using reverse transcriptase was much shorter than that found using other methods. In brief, I suggest that the efficiency of 5' RACE is inversely proportional to the length of poly(A), thereby giving the illusion of a deficiency in capping. The authors need to verify the 5' RACE procedure with RNAs in which the cap is directly preceded by runs of adenylates of varying length. This control was also not done for their earlier papers on yeast VLEs."

The reviewer hypothesized that our results show either that "the efficiency of 5' RACE is inversely proportional to the length of poly(A)" or that "detection of caps is inversely proportional to the length of poly(A)." These statements are thought-provoking, but they are not based on correct assumptions. Our optimized 5' RACE method, which we used for mapping 5' mRNA ends and determining caps, is not based on the work of Schmidt and Mueller (1999). We described the method in detail in our previous work (Vopalensky et al., 2019) and briefly also in this manuscript. We do not use Mn²⁺ in our buffers or ligate adapters. Furthermore, we disagree that we did not provide any method controls in our previous work (Vopalensky et al., 2019). We verified the 5' RACE results of pGKL VLE mRNAs using various biochemical and biological methods. Among these methods, we demonstrated that the in vitro treatment of VLE mRNAs and the control actin mRNA with the decapping enzyme hDcp2, but not Rai1, resulted in the absence of caps at their 5' ends. hDcp2 removes the cap without affecting the rest of the mRNA, including the poly(A) leader. The purpose of this experiment was to test whether VLE mRNAs possess a methylated (m7G) cap at their 5' ends, which was successfully proven. However, the experiment also showed that our 5' RACE method can distinguish precisely between capped and uncapped mRNAs with the same nucleotide sequence. This is because the level of capping was the same for untreated and Rai1-treated mRNAs, while hDcp2-treated mRNAs were mostly detected as uncapped. Additionally, we demonstrated that the production of proteins encoded by uncapped VLE mRNAs is unaffected by the deactivation of the cap-binding translation initiation factor, eIF4E. We also demonstrated that, unlike cellular mRNA represented by HGT1, uncapped VLE mRNA represented by K2ORF5 does not bind to the cap-binding factor eIF4E in vitro. Additionally, we demonstrated differences in the polysome loading and translation of capped and uncapped pGKL VLE mRNA in vivo when the cap-binding eukaryotic translation initiation

factor 4E was inactivated in vivo. All these results agreed with data obtained from 5' RACE and reflected presence or absence of m7G cap on VLE and cellular mRNAs as detected by this method.

However, as I mentioned above, Reviewer 1's comments were substantial. We considered how to provide even stronger evidence for the performance of our 5' RACE method. We decided to use synthetic RNA oligonucleotides that differ only in their 5' end structures and that contain a long 5' poly(A17) leader. One RNA oligonucleotide contained a 5'-ppp-A end to mimic the possible natural structure of uncapped mRNA without additional modification of the 5' end, and the other was terminated with a classical 5' m7GpppA cap. This proved to be an unexpectedly difficult and expensive task. We found only one company in the world that could chemically synthesize these twin RNA oligonucleotides with reasonable quality and length. Our 5' RACE method clearly showed no guanosine cap at the 5' end of all the tested 5'-pppA-terminated RNA oligonucleotides. We detected a cap on approximately 80% of the capped RNA oligonucleotides tested. We assume that the missing caps are probably caused by incomplete modification of the RNA oligonucleotide or by loss of the m7G cap during chemical synthesis, subsequent purification, or storage. In our previous experiments, we did not observe any uncapped transcripts in cellular mRNA represented by the actin mRNA control (Vopalensky et al., 2019). Similarly, we did not detect any uncapped transcripts from VACV early genes without an INR region (see Figures 2 and S1 and Tables S2a and S2b). The 5' RACE method slightly overestimates the length of long 5' homopolymeric poly(A) leaders. The median length of the 5' poly(A) leader detected was 21 adenosines and was the same for both RNA oligonucleotides. We hypothesize that this extension is due to common SuperScript III reverse transcriptase slippage. We did not detect any non-A nucleotides inserted in the poly(A) leader, nor did we detect any other nucleotide extension except for a single G in the case of the capped RNA oligonucleotide. Overall, these results confirm the reliability and precision of the 5' RACE method used.

The 5' RACE method is central to the entire manuscript. In this case, it is difficult to reference all the changes by page/line numbers. The main sections of the manuscript concerning experiments with chemically synthesized RNA oligonucleotides are as follows: pages 6–7 / lines 18–11; pages 17–18 / lines 22–7; and figures 2 (page 21) and S4 (supplemental pages 14–15).

Comment 2: *“The authors never discuss the structure of the putative uncapped RNAs. Presumably, the RNAs have 5' triphosphates. This should be demonstrated and there is a simple way to do this. RNAs with 5' tri- or diphosphates can be capped in vitro with the vaccinia virus capping enzyme, which is available commercially. Indeed, diphosphate ended poly(A) was used as the acceptor substrate in the original paper on purification of capping enzyme (Martin et al. 1975). If the model of the authors is correct, in vitro capping of the RNAs should increase the percentage of capped RNAs detected by the 5' RACE method. If, on the other hand, the RNAs are fully capped, then removal of the caps by periodate oxidation and beta elimination or enzymatically would leave a diphosphate end that could be capped in vitro as shown by Ahn and Moss (1989).”*

As requested by Reviewer 1, we conducted an experiment to shed light on the structure of the 5' ends of uncapped VACV mRNA. We used a well-established method involving the combined treatment of total RNA by RNA 5' polyphosphatase and Terminator 5'-phosphate-dependent exonuclease. This method reliably detects whether the 5' mRNA ends are blocked (e.g., by an m7G cap) and whether they consist of either 5' mono-, di-, or triphosphate. Another advantage is that this method does not use 5' RACE at any stage, providing thus independent information to another results. Our results demonstrate that early VACV mRNAs (represented by J6R transcripts) and some intermediate VACV mRNAs (represented by A2L transcripts) have blocked 5' ends, likely due to an m7G cap. However, late VACV transcripts (represented by C3L transcripts) rarely have blocked 5' ends and instead possess the expected some phosphates at the 5' ends, that means either mono-, di-, or triphosphate ends.

These results are referenced primarily on pages 7–8, lines 12–2; Figure 5 (page 24); and page 18, lines 8–21.

Comment 3: *“A second method, oligo capping (RLM-RACE), was also used to analyze the RNAs of the post-replicative C3L and A17L genes. The lengths of the poly(A) determined by RLM-RACE were compared to that produced by RACE. In the case of A17L, the length of the poly(A) determined for the capped RNA by RLM-RACE was not statistically different from the total RNA determined by RACE. This contradicts their finding that little or no capped A17L RNAs were detected by the 5' RACE method. A statistical difference of $p < 0.01$ was found for the C7L RNA.”*

As Reviewer 1 noted, we employed an independent method to determine whether we could detect 5' capped mRNAs among late VACV mRNAs (represented by A17L and C3L), despite their low proportion (as illustrated in Fig. 5). The rationale behind this experiment was to test whether late mRNAs also support our general observation that 5' capped VACV mRNAs contain shorter 5' poly(A) leaders. We proved this for both genes tested. Currently, box plots are one of the recommended ways to present data because they directly show the overall distribution of the data, including variability, the median, the mean, the range, etc. Frequency statistics is a commonly used tool that should not be overestimated. In our case, the distribution of the observed data clearly shows that the capped mRNA of the two late genes tested (A17L and C3L) has shorter 5' poly(A) leaders than the non-capped mRNA transcribed from the same genes. Additionally, we rejected the null hypothesis for C3L transcripts at the cut-off value below 0.01, which is substantially better than the common threshold < 0.05 .

In conclusion, there is no contradiction with our previous findings regarding uncapped A17L mRNA. Rather, the experiment confirms our previous data showing that the length of nontemplated 5' poly(A) leaders and 5' cap occurrence are negatively correlated. Note that the RLM-RACE method (oligo capping), like many other methods used in previously published experiments to map the 5' ends of VACV mRNAs, relies on heavy and selective amplification of cap-containing mRNAs. This, of course, blurs the presence of uncapped mRNA and cannot help analyzing the proportion of capped and uncapped transcripts of the same gene.

These results are referenced primarily on page 8, lines 3–22; Fig. 5 (page 24); Fig. 6 (page 25).

Comment 4: *“The model that purports to explain decreased capping on pages 9 and 10 and figures 5 and 6 is built on an erroneous supposition that intermediate and late genes are transcribed on an RNA polymerase complex that contains the early transcription factor (VETF) and associated RAP94. There are no data suggesting that intermediate and late transcription occur on an RNA polymerase complex that contains either VETF or RAP94. In fact, VETF and RAP94 are expressed following DNA replication and intermediate and late transcription can occur under conditions in which DNA replication is prevented (Keck et al. 1990). Furthermore, intermediate and late transcription factors as well as the capping enzyme and cap methyltransferase are associated with the vaccinia virus replisome isolated by iPOND, but neither VETF nor RAP94 are present (Senkevich et al. 2017). Belatedly, the authors acknowledge that a different RNA polymerase complex may be used for intermediate and late transcription. Pages 9 and 10 and figures 5 and 6 and subsequent discussion should all be deleted as they will only confuse a reader.”*

I believe I addressed this issue in the introduction of this letter. We agree that the results of molecular dynamics are based on speculative data and assumptions. Therefore, although the results made sense to us, we agree with Reviewer 1's suggestion and have completely deleted the MD results and the corresponding discussion.

Comment 5: *“p. 5 Lines 6 - 7. Statement that “the presence of the 5' cap structure in individual VACV transcripts has never been tested” is incorrect.*

A. Ahn and Moss (1989) analyzed total late mRNA and specifically the late F18 (11 kDa) mRNA. They used a decapping- radioactive recapping method and provided evidence that F18 and all late mRNAs are capped supporting the original report of Boone and Moss (1977).

B. Scher and Stunnenberg (1987) proved that the late 11K (F18L) and 4b (A3L) mRNAs with poly(A) leaders have caps by using anti-cap antibody to immunoprecipitate cDNA-RNA hybrids. In that paper they also showed a discrepancy in the determined length of the poly(A) when reverse transcriptase was used in which it appeared to consist of only 5 to 10 adenylates whereas the length was about 35 adenylates using other methods. The difference was attributed to inefficiency of reverse transcriptase to copy a poly(A) template.

C. Yang et al (2011) showed by CAGE methods that early mRNAs, including rare ones with 5' poly(A), have caps. In another paper Yang et al (2012) identified caps on ~100 individual intermediate and late mRNAs followed by poly(A) by CAGE. These data included the four genes analyzed in the present study: G8R, A1L, C3L and A17L.”

I'd like to thank Reviewer 1 for the important and helpful comment. I reread all the suggested papers and recalled these interesting results. However, there was a misunderstanding. To our knowledge, all previous experiments, including those recommended in the review, analyzed mRNA collectively. That means a

pool of mRNAs transcribed either from a single gene, more genes or even all vaccinia genes together. On the contrary, we can infer the state of a single RNA molecule and calculate the proportion of capped and uncapped mRNA molecules transcribed from a single gene. This process is technically demanding; however, it provided us with a better understanding of the VACV transcript population and VACV transcription in general. I understand that the sentence could be misleading; therefore, we have changed this part of the manuscript as follows:

“However, the presence of the 5' cap structure at individual VACV transcripts has never been tested. It is technically demanding to analyze presence of m⁷G at the individual transcripts. To our knowledge, previous analyses of m⁷G presence at VACV mRNAs evaluated collectively all transcripts from a single gene or group of genes or were biased by preferential selection of capped mRNAs. To achieve the goal of m⁷G cap analysis at individual VACV mRNAs, we decided to apply a modification of the rapid amplification of 5' cDNA ends (5' RACE) method that allows to specifically read the sequence at the 5' mRNA ends including m⁷G cap detection.”

(page 5, lines 9–16)

Comment 6: *“The title is misleading - based on their data it should be “Transcripts of vaccinia virus postreplicative genes may lack a 5' methylguanosine cap”. It should be noted that each of the four intermediate and late genes analyzed in this study were shown to have caps either by the 5' RACE or RLM-RACE method.”*

We followed all of the reviewers' recommendations and performed new experiments to confirm the performance and precision of our 5' RACE method. The 5' RACE results (Fig. 1, Fig. 2, Fig. 4, Fig. 8, Fig. S3, Tables S2a, S2b, S3) and the results from the independent method (Fig. 5) clearly show that the vast majority of VACV late mRNAs are uncapped. Therefore, we believe that the suggested title of the manuscript is correct.

Comment 7: *“p.6, lines 8 - 12. The experiments of Cantu et al are more complex than indicated and would be difficult for the reader to judge. In the absence of decapping enzymes, large amounts of double-stranded RNA accumulate, and KO of RNase L and PKR only partially restores replication suggesting additional host defenses as the simplest explanation. These sentences should be deleted”*

We separated the Results and Discussion chapters as requested. We also rewrote the section on the role of the D9 and D10 decapping enzymes. One reason is that, at the request of Reviewer 2, we now have data from cells infected with the VACV vD9muD10mu strain, which lacks active D9 and D10 decapping enzymes. In light of our new results, we hypothesize that one function of D9/D10 enzymes could be to reduce the proportion of capped cellular mRNAs that compete with viral uncapped transcripts for the cellular translation apparatus. At the same time, they could produce decoy uncapped mRNAs for mRNA degradation pathways, which may increase the lifetime of uncapped viral postreplicative mRNAs.

The production of decoy uncapped mRNAs is commonly considered an important side effect of cap snatching, which is used by some viruses for capped mRNA synthesis. It is also a side effect of the deactivation of host translation factors that are important for the cap-dependent translation pathway, which some viruses use for alternative translation initiation, including IRES-mediated translation. Cantu et al. (2020) specifically demonstrated that the translation of VACV late and 5' polyadenylated mRNA was selectively impaired in vD9muD10mu-infected cells compared to wild-type VACV. Deletion of PKR and RNase L had little effect on this phenomenon. We discuss a possible explanation for Cantu's results. As with some other unexplained published results concerning the translation of late VACV mRNAs, accepting that postreplicative VACV mRNAs are uncapped may often help to provide a solution.

Results and discussion corresponding to this topic are referenced primarily on page 10, lines 5–19; pages 11–12, lines 19–10 and pages 13–14, lines 17–14.

Comment 8: *“P8, lines 20 and Fig. 4. Should point out that Baldick et al. (1992) showed that there was about ~98% inhibition of transcription when one of the A residues in the TAAAAT sequence was mutated and disrupted the AAA initiator indicating a profound effect on the transcription mechanism.”*

Thank you for your comment. We discuss the results of Baldick et al. (1992) in the Discussion chapter now: pages 14–15, lines 15–11.

Comment 9: *“The Results and Discussion should be separated to better distinguish experimental observations and interpretation”*

As recommended, we have separated the Results and Discussion into independent chapters.

Reviewer 2

Introductory paragraph: *“The authors characterized the vaccinia virus (VACV) transcripts at the individual mRNA molecule level and found that some vaccinia virus postreplicative mRNAs, containing non-templated 5' poly(A) leaders, lack the 5' cap structure in vivo. They showed for the late gene G8R that the lengths of the non-templated leaders and the presence or absence of cap structures at the 5' mRNA ends was controlled by the initiator sequence of the VACV postreplicative promoter. With molecular dynamics simulation the authors tried to support a possible link between structural features of the VACV transcribing complex and synthesis of the uncapped poly(A) mRNA leaders.”*

Comment 1: *“The study shows novel aspects of poxviral transcription, but should be supported by additional experiments. The authors analyzed five early genes, two intermediate and two late genes. Even within the early gene mRNAs, the 5' cap occurrence varied significantly. In addition, the intermediate and late mRNAs had different amounts of 5' cap structures. These analyses should be*

performed with more genes to allow for a proper statistical analysis. There are 118 early genes and 93 post replicative genes, more than 5 or 4 each should be analyzed.”

As requested, we added more genes to the analysis. It should be noted that this analysis allows us to finally infer the situation at the single-molecule level, but it is technically demanding and time-consuming. By adding more genes, we were able to separate early genes into two classes: early genes with an INR and early genes without an INR. The latter are more related to intermediate genes in terms of the presence / absence of untemplated 5' poly(A) leaders and 5' caps. We also extended our analysis to include a control consisting of chemically synthesized RNA oligonucleotides with long 5' poly(A) leaders and either 5' pppA or 5' m7GpppA moieties (Figures 2 and S4).

Altogether, we analyzed individual cDNA clones corresponding to VACV transcripts from four early genes without an INR region, three early genes with an INR region, four intermediate genes, and four late genes (Figure 2). In addition, we analyzed transcripts from cells infected with the vD9muD10mu double mutant VACV strain (two early genes, two intermediate genes, and four late genes) (Fig. 8). In total, we sequenced and analyzed the 5' ends of 491 individual cDNA clones.

These results are referenced primarily on Fig. 2 (page 21); Fig. 8 (page 28); Supplementary material pages 2–13 and pages 20–21.

Comment 2: *“The authors claim that the absence of the 5'-cap structure is due to the transcription process and support this theory by an experiment using a mutated promoter sequence of the G8R gene (G8RPM). The mutated promoter sequence resulted in more capped mRNA and less 5'-polyA. The authors should prove that the G8RPM is still transcribed at late time points and should use additional genes to support these data. The data were supported by molecular dynamics simulations using the A17L gene. Why not the G8R?”*

At the request/comments of the other reviewers, we deleted the molecular dynamics experiments. Regarding the G8R experiment, our goal was not to study the transcriptional activity of the mutated G8R promoter. Baldick et al. (1992) thoroughly studied this, preparing and analyzing more than 140 point mutations within the G8R promoter. They also did not observe any difference in the function of the mutated promoter related to the presence or absence of the VACV DNA replication inhibitor, cytosine arabinoside (AraC), except for an overall increase in the expression of all G8R constructs in the presence of AraC. Baldick et al. (1992) and others (Davison and Moss, 1989) also demonstrated that mutations in the INR region are essential for vaccinia promoter function. All published attempts to introduce point mutations in the INR regions of VACV promoters have resulted in weak or abolished expression. In contrast, Ink et al. (1990) showed that shortening the INR sequence of the 38K cowpox promoter inserted into the recombinant VACV did not abolish promoter function, only the formation of the poly(A) leader. We discuss these mutational studies of the INR region in the Discussion chapter. It should be noted that all of the published mutational studies were conducted before methods based on cDNA PCR amplification

became widely used. We benefit from contemporary methods and highly effective engineered enzymes. We detected transcripts from mutated promoters using RT-PCR being a part of the 5' RACE analysis. This analysis was thus successful thanks to cDNA amplification. We also attempted 5' RACE analysis of possible transcripts from the construct with a TACCAT double mutation in the INR region of the G8R promoter, but it was unsuccessful. The latter attempt was unsuccessful, likely due to the impaired function of the G8R promoter. We are discussing all of this in the Discussion section now.

Results and discussion corresponding to this topic are referenced primarily on pages 9–10, lines 1–24 and 1–8; pages 14–15, lines 15–11; Fig. 7 (pages 26–27); pages 18–19, lines 23–5 (plasmid construction); Suppl. Material pages 26–27.

Comment 3: *“VACV is encoding two decapping enzymes; D9 expressed early and D10 expressed late. How can the authors exclude that the reduced amount of 5'-cap structures is not due to decapping, followed by RNA degradation, which would explain the lower amount of 5'-As. Experiments with D9/D10 deleted viruses or inactivated D9/D10 should be performed. In addition, cellular decapping enzymes might be active. Please consider for control experiments.”*

This is a very good comment. We obtained the VACV vD9muD10mu strain, which lacks active D9 and D10 decapping enzymes. This strain was kindly provided to us by Bernard Moss from the NIH/NIAID. 5' RACE analysis of the early, intermediate, and late transcripts of the vD9muD10mu strain revealed no significant differences in the lengths of the 5' poly(A) leaders and the occurrence of the 5' m7G cap between the mutated and wild-type viruses (Figure 8). The shape of the negative correlation between the number of nontemplated adenosines in the 5' mRNA leader and 5' cap occurrence is similar in the wild-type (wt) and vD9muD10mu strains (Figure 4).

Results and discussion corresponding to this topic are referenced primarily on page 10–11, lines 8–1; Fig. 4 (page 23); Fig. 8 (page 28); pages 13–14, lines 17–6; Pages 16–17, lines 13–6 (method); Suppl. Material pages 3–4, 6–7, 9–13, 21, 23.

Reviewer 3:

Comments: *“Poxviruses are large cytosolic DNA viruses that encode a viral multi-subunit DNA-dependent RNAPolymerase (vRNAP). Members of this family are responsible for a wide range of relevant infectious diseases in humans and animals, including monkeypox and smallpox. A thorough understanding of their unique cytosolic transcription apparatus is of general interest, not only since some of its components are widely used as a biotechnical tool (in particular the vaccinia capping enzyme). Until recently, it was a widespread belief that vaccinia transcripts are heavily modified with a cap structure at their 5' end. Here, for the first time, Vopálenský et al. use RACE PCR to examine the 5' ends of vaccinia early, intermediate and late transcripts with the surprising finding that only roughly half of the early transcripts are capped and most of the intermediate and late transcripts are not capped at all. They further provide convincing experimental evidence that the*

occurrence of a cap structure is strictly negatively correlated to the appearance of non-encoded poly-A stretches in the 5'-UTRs. Finally, a "slipping" mechanism is discussed that probably is an explanation for the generation of non-encoded poly-A stretches by vRNAP. This hypothesis is backed by a molecular dynamics (MD) simulation of a respective DNA/RNA hybrid in the range of the INR promoter element. The experimental methods used in this study appear technically sound and the results are of genuine scientific interest. The manuscript therefore deserves publication after the following shortcomings have been corrected:

1. The MD simulation of the DNA/RNA hybrid results in the transformation of the double helix into an extended 'ladder'. A comparative simulation with a corresponding hybrid of a non-slipping promoter should be performed and the results presented to validate the simulation. Please describe also in the methods section, how the hybrid coordinates were generated.

2. The results of the MD simulation can only be validly discussed in the context of the vRNAP transcription machinery if the ladder conformation can be reasonably fitted in the active site cleft. A respective figure should be presented in the supplemental section.

3. The authors argue that for intermediate/late transcripts, the "Nascent 5' end of the viral mRNA thus cannot efficiently displace Rap94 B-reader element and enter the RNA exit tunnel leading to the TPase active site. This situation leads to synthesis of mRNAs with longer nontemplated 5' poly(A) 20 leaders which exit vRNAP by different way than viral early mRNAs and thus can escape the viral capping machinery" (p. 10, l. 17ff). This notion is problematic as the current understanding is that vaccinia intermediate and late transcription is not dependent on Rap94 (i. e. one would expect the exact opposite: that early transcripts frequently slip and intermediate/late do not).

Minor points:

- p. 16, l. 5: This seems to refer to the Amber ff99SB-ILDN force field.

- p. 9, l. 13: The nascent mRNA strand ...

- p. 10, l. 22: "However, we also cannot exclude a possibility" should read: "However, we also cannot exclude the possibility"

- p. 11, l. 3: "Nevertheless, only part of the vRNAPs is associated with Rap94" should read: "Nevertheless, only a part of the vRNAP population is associated with Rap94"

Thank you very much for the positive review and useful comments. After considering all the reviews, particularly those of Reviewer 1, we have decided to remove the MD part from the manuscript. However, we will continue in this direction, and your comments will be very useful for our future work. Therefore, I will not address your comment regarding the MD results and the Rap94-related hypotheses, which were also removed at Reviewer 1's request. We expanded our results considerably by including 5' RACE analyses from more genes using both the wild-type virus and a double mutant vD9muD10mu with both

viral decapping enzymes rendered inactive. We also included a direct biochemical analysis of VACV 5' mRNA ends.

Thank you all for considering our manuscript and helping us improve it.

Sincerely yours,

Martin Pospisek

Dear Reviewers,

Thank you for your time and help with improving our manuscript.

Answers to the reviewers' comments:

Reviewer 1

1] We changed the title and abstract so that we eliminated the reference to "postreplicative" mRNA and instead pointed out that 5' polyadenylated mRNA is uncapped. However, we did not extend this definition to "long" poly(A) leaders. The prefix "poly-" means "many" monomer units anyway. The combination of "poly" and "long" might mislead the reader into thinking that only the very long 5' leaders are uncapped. In fact, Fig. 4 shows that, starting at five to six adenosines in length, there is almost no capping.

The new title is as follows:

"Vaccinia Virus mRNAs Containing 5' Polyadenosine Leaders Lack a Methylguanosine Cap"

2] We added a brief description of the 5' RACE method to the Results section as requested. This part of the manuscript is as follows:

To achieve the goal of m⁷G cap analysis at individual VACV mRNAs, we decided to apply a modification of the Rapid Amplification of 5' cDNA Ends (5' RACE) method that allows to specifically read the sequence at the 5' mRNA ends including m⁷G cap detection. In brief, the method takes advantage of SuperScript III reverse transcriptase's ability to bypass the 5'–5' triphosphate bond of the m⁷GpppN cap and insert cytosine into the cDNA position, which is complementary to the 5' methylguanosine cap moiety. The cDNA was then tailed using terminal deoxynucleotidyl transferase and dGTP. The 5' mRNA ends were inferred upon amplification using the cDNA as a template, an oligo(dC) anchor primer, and a gene-specific primer. Resulting amplicons were cloned into the pCR4-TOPO vector and sequenced. To eliminate amplification bias, clones from several independent amplification rounds were used for the analysis.

3] I understand that knowing the number of phosphates at the 5' ends would be helpful. However, it is not that easy to perform the corresponding experiments in a well-controlled manner. We are working on it and hope to publish more details in the future. To inform the reader of the current state of our research, we mention that: "This result can be interpreted that while *J6R* mRNAs and part of *A2L* mRNAs have blocked 5' ends e.g. by m⁷G caps, *C3L* mRNAs possess unblocked 5' tri-, di- or monophosphate ends."

Thus, readers are informed that the number of phosphates at the 5' ends of late mRNAs may differ.

4] We added the following sentence to the appropriate place in the Discussion section: "Although published data clearly suggests that vaccinia virus uncapped and 5' polyadenylated mRNAs could be

loaded onto polysomes and translated by an eIF4E-independent pathway, these experiments have yet to be performed. We are currently working in this direction.“

Sincerely yours,

Martin Pospisek
